# Impact of taxes and warning labels on red meat purchases among US consumers: A randomized controlled trial

Lindsey Smith Taillie[1,2]*, Maxime Bercholz[1], Carmen E. Prestemon[1], Isabella C. A. Higgins[1,3], Anna H. Grummon[4], Marissa G. Hall[1,3,5], Lindsay M. Jaacks[6]

1 Carolina Population Center, University of North Carolina, Chapel Hill, North Carolina, United States of America, 2 Department of Nutrition, Gillings School of Global Public Health, University of North Carolina, Chapel Hill, North Carolina, United States of America, 3 Department of Health Behavior, Gillings School of Global Public Health, University of North Carolina, Chapel Hill, North Carolina, United States of America, 4 Department of Pediatrics, Stanford University School of Medicine, Palo Alto, California, United States of America, 5 Lineberger Comprehensive Cancer Center, University of North Carolina, Chapel Hill, North Carolina, United States of America, 6 Global Academy of Agriculture and Food Systems, University of Edinburgh, Midlothian, United Kingdom

* taillie@unc.edu

**Data Availability Statement:** All data files are available from the Harvard Dataverse database (https://doi.org/10.7910/DVN/HQMFUS).

## Abstract

### Background

Policies to reduce red meat intake are important for mitigating climate change and improving public health. We tested the impact of taxes and warning labels on red meat purchases in the United States. The main study question was, will taxes and warning labels reduce red meat purchases?

### Methods and findings

We recruited 3,518 US adults to participate in a shopping task in a naturalistic online grocery store from October 18, 2021 to October 28, 2021. Participants were randomized to one of 4 conditions: control (no tax or warning labels, $n = 887$), warning labels (health and environmental warning labels appeared next to products containing red meat, $n = 891$), tax (products containing red meat were subject to a 30% price increase, $n = 874$), or combined warning labels + tax ($n = 866$). We used fractional probit and Poisson regression models to assess the co-primary outcomes, percent, and count of red meat purchases, and linear regression to assess the secondary outcomes of nutrients purchased. Most participants identified as women, consumed red meat 2 or more times per week, and reported doing all of their household's grocery shopping. The warning, tax, and combined conditions led to lower percent of red meat–containing items purchased, with 39% (95% confidence interval (CI) [38%, 40%]) of control participants' purchases containing red meat, compared to 36% (95% CI [35%, 37%], $p = 0.001$) of warning participants, 34% (95% CI [33%, 35%], $p < 0.001$) of tax participants, and 31% (95% CI [30%, 32%], $p < 0.001$) of combined participants. A similar pattern was observed for count of red meat items. Compared to the control, the combined condition reduced calories purchased (−311.9 kcals, 95% CI [−589.1 kcals,

**Funding:** Wellcome Trust, grant id #216042/Z/19/Z. K01HL147713 and K01HL158608 from the National Heart, Lung, and Blood Institute of the NIH supported MGH's and AHG's time, respectively. ICAH received support from the NICHD-NRSA Population Research Training grant (T32 HD007168). The content is solely the responsibility of the authors and does not necessarily represent the official views of the NIH. The funders had no role in study design, data collection and analysis, decision to publish, or preparation of the manuscript.

**Competing interests:** I have read the journal's policy and the authors of this manuscript have the following competing interests: ICAH purchased some stock in Beyond Meat. The author made this purchase prior to knowing the study results. The other authors have declared that no competing interests exist.

−34.7 kcals], $p = 0.027$), while the tax (−10.3 g, 95% CI [−18.1 g, −2.5 g], $p = 0.01$) and combined (−12.7 g, 95% CI [−20.6 g, −4.9 g], $p = 0.001$) conditions reduced saturated fat purchases; no condition affected sodium purchases. Warning labels decreased the perceived healthfulness and environmental sustainability of red meat, while taxes increased perceived cost. The main limitations were that the study differed in sociodemographic characteristics from the US population, and only about 30% to 40% of the US population shops for groceries online.

## Conclusions

Warning labels and taxes reduced red meat purchases in a naturalistic online grocery store.

**Trial Registration:** http://www.clinicaltrials.gov/ NCT04716010.

---

## Author summary

### Why was this study done?

- A large body of research from tobacco, alcohol, and sugar-sweetened beverages finds that point-of-purchase policies such as warning labels and taxes reduce purchases of these products.

- However, little is known about whether these policies would similarly reduce purchases of red meat, an important contributor to noncommunicable disease risk and environmental damage in the United States.

### What did this study do and find?

- This study evaluated the effects of health and environmental warning labels and taxes on red meat purchases using a randomized controlled trial in a realistic online supermarket ($n = 3,518$ US adults).

- Warning labels, taxes, and a combination of warning labels and taxes all led to modest but statistically significant reductions in red meat purchases compared to a no-policy control.

- The condition that included both warning labels and taxes had the largest impact on red meat purchases and also reduced calories and saturated fat purchased.

- Warning labels led to lower perceived healthfulness and environmental sustainability of red meat, while taxes led to higher perceived cost.

- The policies' impact on purchases was greatest for lower-educated and younger participants.

### What do these findings mean?

- Implementing health and environmental warning labels and taxes could reduce red meat purchases in the US, ultimately providing health and environmental co-benefits.

- The main limitations were that the study differed in sociodemographic characteristics from the US population, and only about 30 to 40% of the US population shops for groceries online.

## Introduction

Red meat consumption negatively impacts both the environment and consumers' health. Red meat production is a major contributor to global greenhouse gas emissions, land clearing, and biodiversity loss, and red meat consumption has been found to increase the risk of several non-communicable diseases including type 2 diabetes, cardiovascular disease, and colorectal cancer [1]. The United States is among the world's top producers of greenhouse gas emissions and top consumers of red meat, with nearly half of Americans consuming red meat on a given day [2]. However, a recent survey found that approximately 70% of Americans are concerned about addressing global climate change [3], and another survey found that 40% of Americans were trying to reduce their red meat consumption [4], signaling that implementing population-wide interventions to reduce red meat consumption may be possible.

Public policies are a critical tool for reducing red meat production and consumption because they can affect entire populations and stimulate changes in supply and demand. The landmark 2019 EAT-*Lancet* Commission report noted that a full range of population-level interventions—from "soft" interventions such as consumer awareness campaigns to "hard" interventions such as fiscal measures—will be essential to transform food systems and diets to achieve targets for health and sustainability [5]. Warning labels and taxes are 2 policies that have been effectively used to reduce consumption of tobacco, sugar-sweetened beverages, and other unhealthy foods [6,7]. Modeling studies have found that meat taxes would reduce consumption and lead to reduced mortality and healthcare costs, increased quality of life, and higher productivity [8–14]. Similarly, one previous study evaluated graphic warning labels on meat products and found that a disgust-eliciting image under the guise of trivia reduced participants' intention to consume meat compared to an explicit warning label [15]. Little experimental research, however, has examined how these policies affect purchases of red meat, and to our knowledge, no policies have been implemented to target red meat reduction.

Empirical data in this area are important because it is unclear whether taxes and warning labels on red meat would have similar effects as they have on other products like sugary drinks, tobacco, and alcohol. In the US, meat is typically consumed as the main part of a meal. Because of its more central role in the American food basket, it is possible that consumers are less likely to view red meat products as "discretionary" or "nonessential" than they would with sugary drinks, and, thus, they may be less likely to change their behavior in response to taxes or labels. On the other hand, relative to sugary drinks, where the majority of Americans are aware of common health harms associated with consumption [16], a much smaller proportion are aware of the harms associated with red meat [17]. This suggests that warning labels that educate consumers about these health and environmental harms may reduce consumption.

Health behavior and economic theory and research indicate that these policies could lead to reductions in purchases through multiple pathways. Meat taxes reduce consumption by increasing prices and, thus, reducing demand for these products [18]. Warning labels reduce consumption by increasing attention, helping people understand the healthfulness or the environmental impacts associated with the product, increasing risk perceptions about harms associated with consumption, and changing behavioral intentions, leading to reductions in selecting meat products for purchase [19].

Although prior research suggests that warnings and taxes reduce purchases of products like sugary drinks, tobacco, and alcohol, it is not clear whether these policies will have similar effects on red meat. In the US, meat is typically consumed as the main part of a meal [20]. Because of its more central role in the American food basket, it is possible that consumers are less likely to view red meat products as "discretionary" or "nonessential" than they would with sugary drinks, tobacco, or alcohol, and thus they may be less likely to change their behavior in response to taxes or labels. On the other hand, relative to sugary drinks, where the majority of Americans are aware of common health harms associated with consumption [16], a much smaller proportion are aware of the harms associated with red meat [17]. This suggests that warning labels that educate consumers about health and environmental harms may reduce consumption.

To address these gaps and inform policymaking, the main study question was, will taxes and warning labels reduce red meat purchases? To address this question, we tested the impact of taxes, warning labels, and taxes combined with warning labels on red meat purchases among US red meat consumers using a randomized controlled trial in a naturalistic online grocery store. Secondary objectives included testing the impact of these policies on nutrient purchases, psychological outcomes, and policy support.

## Methods

### Ethics statement

All study procedures were reviewed and approved by The University of North Carolina at Chapel Hill Institutional Review Board (IRB #19–3349). All participants provided online written informed consent.

### Experimental setting

The trial took place in a naturalistic online grocery store developed for use in research studies [21]. Briefly, we created an online shopping platform designed to mirror the top food retailer (by market share) in the US. The store included over 13,000 products (**S1 Appendix**), including the full range of foods and beverages stocked by the store and including all departments and aisles. Participants could navigate the online grocery store using a search bar or by clicking on departments, aisles, and shelves, similar to a real online store. The online grocery store also included a shopping list, shopping cart, and check-out features.

### Participants

A convenience sample of 5,533 participants was recruited from October 18, 2021 to October 28, 2021 by CloudResearch (Prime Research Solutions LLC, New York) to complete an online survey and experiment. CloudResearch targeted recruitment to create a sample that approximately represented the US population in terms of gender, race/ethnicity, income, and age. Participants were eligible if they were 18 years or older, currently resided in the US, reported eating red meat at least 1 or more times per week during the last 30 days, and reported typically doing at least half of the grocery shopping for their household (**S1 Table**). Compared to the US population, our sample had higher representation of women, older adults, people identifying as white and people identifying as not Hispanic, Latino, or Spanish origin (**S2 Table**).

### Design

After providing online written informed consent, participants were randomly assigned to one of 4 conditions in a 1:1:1:1 ratio: control (*n* = 887), warning labels (*n* = 891), tax (*n* = 874), or

combined (warning labels + tax, $n = 866$). In the control condition, products containing red meat retained their original price and did not display any additional labels on the product page. In the warning labels condition, red meat products had health and environmental warning labels applied next to the product image. In the tax condition, prices of red meat products were raised 30% over the baseline price. In the combined condition, red meat products received both the warning labels and the tax.

## Stimuli

We developed one health ("WARNING: Eating red meat increases your risk of colon cancer and rectal cancer") and one environmental ("WARNING: Eating red meat harms the environment") warning (**Fig 1**). The warning labels were designed based on previous research on warning efficacy [19,22] and nutrient warning labels in effect globally [23]. We tested several health and environmental warnings for red meat [24,25] and selected one for each topic (health and environment) based on perceived message effectiveness, evidence for the claims, literacy, and political feasibility through consultation with nutrition and environmental scientists and public health lawyers. The tax level of 30% was determined as a tax level likely to reduce health and environmental harms [8,10].

## Procedures

**Shopping task.** Participants completed a shopping task in the naturalistic online grocery store. Participants selected items using a predetermined 9-item shopping list: 1 pizza, 1 burrito, burger patties (meat or vegetarian), breakfast sausages (meat or vegetarian), 1 frozen individual meal, 1 loaf of bread, 1 sandwich filling (for example, ham, turkey, or peanut butter), 1 pack of tortillas, and 1 taco filling (for example, steak, chicken, or beans). The shopping list was informed by data on the main sources of red meat consumption in the US [2]. Participants were informed that their budget was $40, similar to the amount spent by participants during

**Fig 1. Warning labels used in trial.**

cognitive testing of the store and shopping protocol. To check out of the store, a participant's shopping basket had to have +/−2 items of the total number of items on the shopping list (i.e., 7 to 11 items). To encourage participants to select products that they would purchase in real life, participants were informed that 10% of participants would be randomly selected to receive the groceries they selected during the shopping task and the remainder of their budget in cash. Around 10% of participants were randomly selected to receive a $40 gift card.

After completing the shopping task, participants were directed to a survey to complete a post-shopping questionnaire. All participants were given an incentive paid in points redeemable by CloudResearch for completion of the survey and experiment.

## Measures

**Primary outcomes.**   The co-primary outcomes were the count of products that contained red meat and the percent of products in the shopping basket that contained red meat. We hypothesized that, compared to the control condition, both taxes and warnings would lead to reduced purchases of products that contain red meat, and the combination of taxes and warnings would lead to a larger reduction in purchases of products that contain red meat than taxes or warnings alone.

**Secondary outcomes.**   Secondary behavioral outcomes included the total saturated fat, sodium, and calories purchased, calculated by summing these outcomes across all items in the participants' basket. We additionally conducted exploratory (not preregistered) analyses examining the weight of red meat items purchased, given that policies designed to reduce purchase of an item can do so by reducing the number of items purchased (as operationalized through the count and percentage outcomes) as well as the size or amount of the item purchased (operationalized through the weight of the product).

The study additionally assessed secondary psychological outcomes hypothesized as potential drivers of behavior change. Secondary psychological outcomes were assessed with survey items (S1 Table). All items were assessed on 1 to 5 Likert-type scales, coded as 1 (low) to 5 (high).

These outcomes included cognitive elaboration, or how much they thought about health harms, environmental harms, and price, during the shopping task. Next, the survey assessed perceptions of eating red meat in general, including perceived healthfulness, perceived risk of cancer, and perceived environmental harms. The survey additionally examined intentions to reduce consumption.

Next, the survey assessed perceptions of specific red meat products (burger patties, pepperoni pizza, and ham luncheon meat, shown in random order). Products were shown as they appeared in the participants' trial condition (e.g., participants randomized to the tax condition were shown images of the products with their price including the 30% tax). For each product, the survey assessed perceived healthfulness, perceived sustainability (how bad or good for the environment the product is), and perceived cost. We averaged the perceived healthfulness and the perceived sustainability of the burger, pizza, and ham products into 2 single outcomes (perceived healthfulness and perceived sustainability of specific red meat products), as Cronbach's alpha for these products was greater than 0.7 for both healthfulness and sustainability. However, it was less than 0.7 for the perceived cost of these products, and so these were analyzed as separate outcomes. Reliability of 0.7 is a standard cutoff for demonstrating that items in a scale have sufficient internal consistency [26].

The survey also assessed support for red meat taxes, health warnings, and environmental warnings. For simplicity of interpretation, we created a binary variable for support for each policy (i.e., responses of "agree" or "strongly agree"). We also examined effects on the lack of support for these policies (i.e., responses of "disagree" or "strongly disagree").

Process measures included whether the participant could easily find all food and beverage items they were looking for, whether there were enough options, whether the store felt like a real online grocery store, and whether the items they selected in the store were similar to their usual grocery purchases. Three additional items assessed interest in health and six additional items assessed interest in sustainability. Items for health and for sustainability were each combined into a scale, except for the item "I do not need to change my diet as it is healthy enough" (which we did not use in subsequent analyses), as it resulted in Cronbach's alpha for health interest dropping below 0.7.

Finally, the survey assessed demographic information including age, gender, race/ethnicity, political affiliation, educational attainment, and household income.

## Statistical analysis

Analyses, hypotheses, and data sharing plan were preregistered on www.clinicaltrials.gov (NCT04716010) (**S1 Statistical Analysis Plan** and **S1 Study Protocol**). All statistical analyses were conducted in Stata Statistical Software, Release 17 (StataCorp LLC, College Station, Texas, USA). We used two-tailed tests and a significance level of 0.05. We report unadjusted values for the primary, secondary, and other outcomes. Per CONSORT guidelines, we did not test for balance in covariates across conditions [27].

Sample size calculations were conducted using PASS 2019 Power Analysis and Sample Size Software (NCSS, Kaysville, Utah, USA). The tax was expected to have a larger effect than the warning label, so the study was powered to detect the main effect of the warning labels. The study was originally powered for an analysis of variance (ANOVA) to detect a Cohen's f = 0.05 main effect of the warning labels with 80% power at the 5% level ($n$ = 786 per condition). However, we chose to compare individual conditions instead, and post hoc power calculations showed that we had a large enough sample ($n$ = 866 to 891 per condition) to detect a Cohen's d = 0.13 difference between the warnings and control conditions at the 5% level with 80% power. For reference, given the observed mean count and percent of red meat items in the control condition and standard deviations, a Cohen's f of 0.05 corresponds to effects of approximately 0.25 on the count of red meat items (a 7% difference) and 2.7 percentage points on the percent of red meat items, with similar effects (0.23 and 2.5, respectively) for a Cohen's d of 0.13.

We used regression models to assess differences in the outcomes by condition, comparing each of the 3 intervention conditions (warnings only, taxes only, warnings + taxes) to the control condition. We also assessed differences in the primary outcomes between the 3 intervention conditions. We used Poisson and fractional probit regression models for the count and percent of items in the basket that contained red meat, respectively, and linear regression models for the remaining outcomes, with condition as the independent variable. We used robust standard errors in all models except linear regression models where we did not find evidence of heteroskedastic errors based on Breusch–Pagan test assuming independent and identically distributed (IID) errors. Differences between conditions were assessed by comparing the predicted means by condition.

In order to understand whether taxes and warning labels had a differential effect on red meat purchases for specific sociodemographic subpopulations, we carried out moderation analyses. Specifically, we examined whether intervention effects on the primary outcomes differed by participant characteristics including red meat consumption in the past 30 days, interest in health, interest in sustainability, household income in the last 12 months, education, age, race/ethnicity, political orientation, and gender (**S2 Appendix**). For each of these potential moderators, we fit a Poisson regression model for the count outcome and a fractional probit

model for the fractional outcome with condition, the moderator, and their interaction as the independent variables. Using the model estimates, we then estimated the difference between each intervention condition and the control condition at each level of the moderator and tested the hypothesis of equal differences with the control condition across those levels for each intervention condition, examining all pairwise comparisons (e.g., for age, 18 to 39 years versus 40 to 59 years, 18 to 39 years versus 60 years or over, and 40 to 59 years versus 60 years or over). Prior to analyses, we collapsed several moderators to avoid small cell sizes (S2 Appendix).

As preregistered, we carried out sensitivity analyses for the main comparisons on the primary outcomes. Specifically, we repeated the analysis excluding participants in the bottom 2% of total spending, participants who completed the study in less than half the median completion time of 18 minutes and 46 seconds, and participants who purchased less than half of the items on the shopping list, using each product's department, aisle, and shelf, as well as manual inspection when in doubt, to determine if it was a valid shopping list item. Finally, we examined whether dropping out of the shopping task (i.e., started but did not complete the task) was associated with condition or any of the demographic characteristics collected in the screener and checked the robustness of the results to these associations.

## Results

A total of 3,518 participants completed the shopping task (Fig 2). Sample characteristics are shown in Table 1. Most participants reported doing all of their households' grocery shopping, and more than half the sample reported shopping for groceries online within the last month. About 85% of the sample consumed red meat 2 or more times per week. Descriptive statistics on outcomes can be found in S3 and S4 Tables.

In the control condition, an average of 39% (95% confidence interval [CI]: 38%, 40%) of the products participants selected contained red meat. Relative to the control group, participants in the warning label, tax, and combined conditions purchased a lower percent of red meat items, 36% (95% CI [35%, 37%], $p = 0.001$), 34% (95% CI [33%, 35%], $p < 0.001$), and 31% (95% CI [30%, 32%], $p < 0.001$), respectively (Fig 3, Panel A, and S5 Table). When comparing

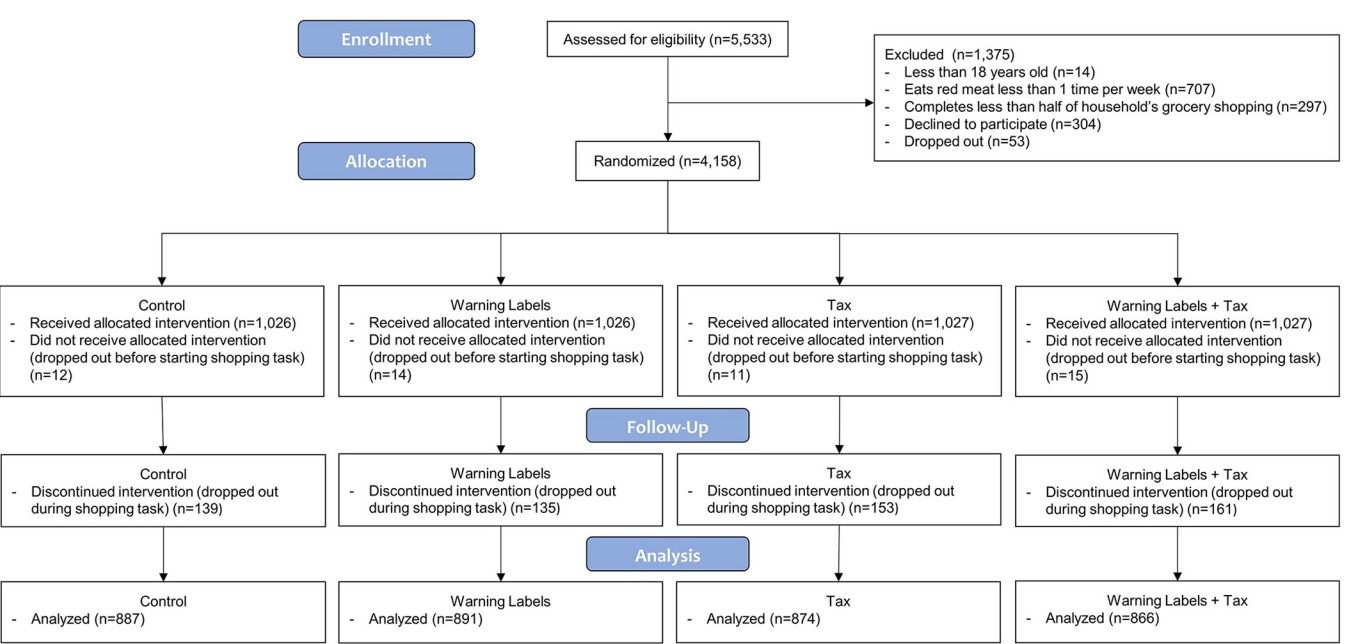

**Fig 2. Participant flow chart (final sample analyzed = 3,518).**

**Table 1. Sociodemographic characteristics by condition (*n* = 3,518).**

| | Control (*n* = 887) | Warning Label (*n* = 891) | Tax (*n* = 874) | Warning Label + Tax (*n* = 866) | All (*n* = 3,518) |
|---|---|---|---|---|---|
| | n (%) | n (%) | n (%) | n (%) | n (%) |
| **Gender** | | | | | |
| Woman | 539 (60.8) | 538 (60.4) | 530 (60.6) | 519 (59.9) | 2,126 (60.4) |
| Man | 343 (38.7) | 348 (39.1) | 342 (39.1) | 344 (39.7) | 1,377 (39.1) |
| Nonbinary or self-described | 5 (0.6) | 5 (0.6) | 2 (0.2) | 3 (0.3) | 15 (0.4) |
| **Age** | | | | | |
| 18–39 | 304 (34.3) | 305 (34.2) | 278 (31.8) | 335 (38.7) | 1,222 (34.7) |
| 40–59 | 302 (34.1) | 283 (31.8) | 283 (32.4) | 273 (31.5) | 1,141 (32.4) |
| 60+ | 281 (31.7) | 303 (34.0) | 313 (35.8) | 258 (29.8) | 1,155 (32.8) |
| **Education** | | | | | |
| High school diploma or less | 292 (33.2) | 311 (35.1) | 291 (33.6) | 294 (34.2) | 1,188 (34.1) |
| Associate or technical degree | 191 (21.7) | 197 (22.3) | 214 (24.7) | 204 (23.7) | 806 (23.1) |
| 4-year college degree | 269 (30.6) | 250 (28.2) | 250 (28.9) | 255 (29.7) | 1,024 (29.3) |
| Graduate degree | 127 (14.4) | 127 (14.4) | 111 (12.8) | 106 (12.3) | 471 (13.5) |
| **Household income** | | | | | |
| Lower ($0 to <$35,000) | 244 (27.8) | 294 (33.2) | 271 (31.3) | 255 (29.7) | 1,064 (30.5) |
| Middle ($35,000 to <$74,999) | 321 (36.6) | 319 (36.1) | 316 (36.5) | 333 (38.8) | 1,289 (37.0) |
| Higher (≥$74,999) | 313 (35.7) | 272 (30.7) | 278 (32.1) | 271 (31.6) | 1,134 (32.5) |
| **Race and ethnicity[a]** | | | | | |
| Hispanic (any race) | 83 (9.4) | 94 (10.6) | 85 (9.8) | 94 (10.9) | 356 (10.2) |
| NH White | 643 (73.2) | 654 (73.8) | 634 (73.2) | 621 (72.3) | 2,552 (73.1) |
| NH Black or African American | 80 (9.1) | 87 (9.8) | 76 (8.8) | 89 (10.4) | 332 (9.5) |
| NH Asian or Pacific Islander | 33 (3.8) | 29 (3.3) | 38 (4.4) | 31 (3.6) | 131 (3.8) |
| NH Other/Multiracial | 40 (4.6) | 22 (2.5) | 33 (3.8) | 24 (2.8) | 119 (3.4) |
| **Red meat consumption** | | | | | |
| 1 time/week | 141 (15.9) | 130 (14.6) | 125 (14.3) | 129 (14.9) | 525 (14.9) |
| 2–3 times/week | 447 (50.4) | 430 (48.3) | 416 (47.6) | 403 (46.5) | 1,696 (48.2) |
| 4–6 times/week | 187 (21.1) | 204 (22.9) | 217 (24.8) | 192 (22.2) | 800 (22.7) |
| ≥1 time/day | 112 (12.6) | 127 (14.3) | 116 (13.3) | 142 (16.4) | 497 (14.1) |
| **Political orientation** | | | | | |
| Liberal | 272 (30.9) | 238 (26.9) | 244 (28.2) | 254 (29.6) | 1,008 (28.9) |
| Moderate | 339 (38.6) | 355 (40.1) | 326 (37.7) | 324 (37.8) | 1,344 (38.5) |
| Conservative | 268 (30.5) | 293 (33.1) | 294 (34.0) | 280 (32.6) | 1,135 (32.5) |
| **Interest in health[b]** | | | | | |
| Low | 93 (10.6) | 104 (11.7) | 95 (11.0) | 93 (10.8) | 385 (11.0) |
| Moderate-low | 233 (26.5) | 231 (26.1) | 252 (29.1) | 231 (26.9) | 947 (27.1) |
| Moderate-high | 432 (49.2) | 447 (50.5) | 409 (47.2) | 444 (51.7) | 1,732 (49.6) |
| High | 121 (13.8) | 104 (11.7) | 110 (12.7) | 91 (10.6) | 426 (12.2) |
| **Interest in sustainability[c]** | | | | | |
| Low | 69 (7.9) | 70 (7.9) | 72 (8.3) | 68 (7.9) | 279 (8.0) |
| Moderate-low | 176 (20.1) | 152 (17.2) | 164 (18.9) | 151 (17.6) | 643 (18.4) |
| Moderate-high | 362 (41.2) | 382 (43.1) | 385 (44.5) | 396 (46.1) | 1,525 (43.7) |
| High | 271 (30.9) | 282 (31.8) | 245 (28.3) | 244 (28.4) | 1,042 (29.9) |

[a]Self-reported identity. NH, non-Hispanic; NH White and NH Black or African American exclude NH multiracial; NH Asian or Pacific Islander include NH Asian only, NH Pacific Islander only, and NH Asian and Pacific Islander.

[b]Based on a scale of self-perceived dietary behavior [28].

[c]Based on the GREEN Scale [29].

**A)**

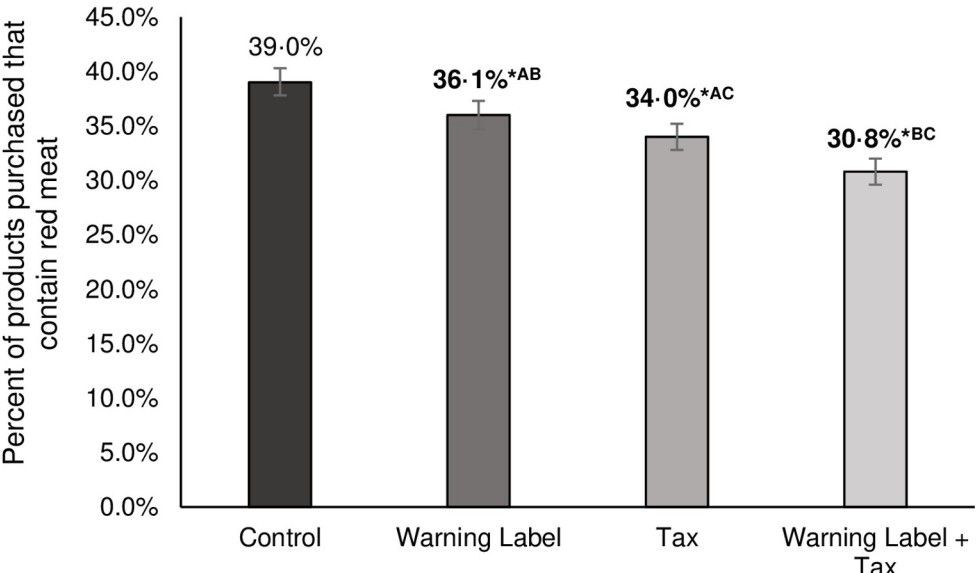

**B)**

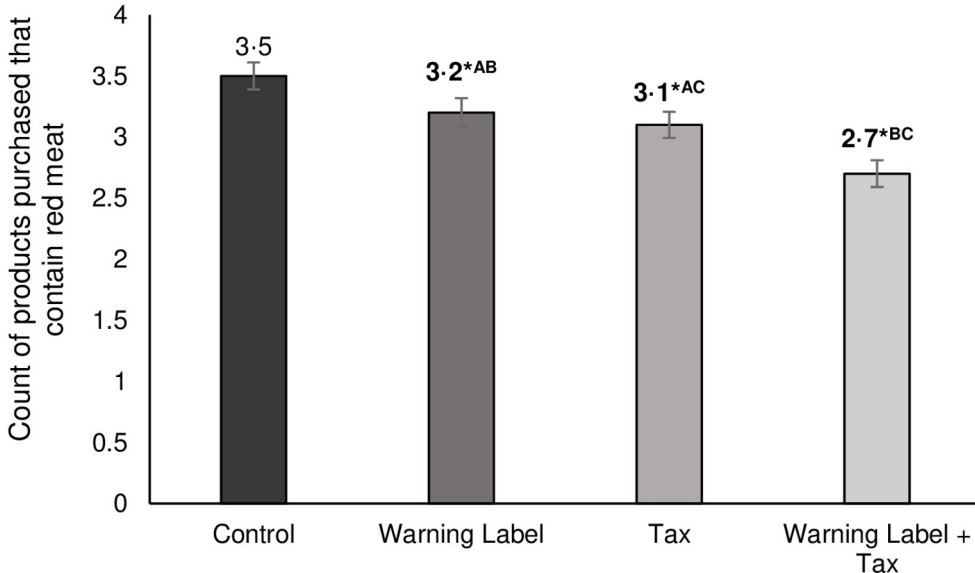

**Fig 3. (A) Percent of red meat items purchased during the shopping task, by condition; (B) count of red meat items purchased during shopping task, by study arm.** *Asterisk denotes statistically significant differences between each intervention and the control at the 5% level. [a]Shared superscript indicates significant difference between interventions at the 5% level. [b]Error bars show 95% confidence interval (CI). [c]P values were calculated using fractional probit regression models for percent and Poisson regression models for count.

differences between conditions, participants in the tax condition purchased a lower percentage of products containing red meat than did people in the warning label condition ($p = 0.022$), and participants in the combined condition purchased a lower percentage of products containing red meat than did participants in either the warning or tax condition ($p < 0.001$).

Results for count outcomes followed a similar pattern. Each condition reduced the number of items containing red meat purchased relative to the control. The average count of purchased items containing red meat in the shopping basket for the control condition was 3.5 items (95% CI [3.4 items, 3.6 items]), compared to 3.2 items (95% CI [3.0 items, 3.5 items]) in the warning label condition ($p = 0.002$) and 3.1 items (95% CI [2.8 items, 3.3 items]) in the tax condition ($p < 0.001$). The combined led to the smallest number of items containing red meat purchased (2.7 items, 95% CI [2.6 items, 2.9 items]) relative to the control, warning label, and tax conditions ($p < 0.001$ for all 3 comparisons) (**Fig 3, Panel B,** and **S5 Table**).

The impact of the combined condition on red meat purchases was moderated by education and age (**S6** and **S7 Tables**). Relative to the control, the combined condition led to fewer red meat purchases for all education levels except those with a graduate degree. For example, among participants who had a high school degree, relative to the control, the combined condition led to an 8.7 percentage point reduction in red meat purchases (95% CI [−11.6 percentage points, −5.7 percentage points]). In contrast, among those with a graduate degree, the combined condition had no impact on red meat purchases. Additionally, relative to the control, the combined condition led to larger reductions in red meat purchases for the youngest age group (−11.0 percentage points, 95% CI [−13.8 percentage points, −8.3 percentage points]) than for the oldest age group (−5.0 percentage points, 95% CI [−8.2 percentage points, −1.8 percentage points]). There was no evidence of moderation for the other conditions or by other variables, and the pattern of moderation results for count of red meat purchases was similar (**S6** and **S7 Tables**).

The conditions also affected secondary behavioral outcomes, with effects varying by nutrient (**Table 2**). Relative to the control, the combined condition led to fewer calories purchased (-311.9 calories, 95% CI [-589.1 calories, -34.7 calories], $p = 0.027$). Both the tax and combined conditions led to lower saturated fat purchased compared to control ($p = 0.01$ and $p = 0.001$, respectively), as well as reductions in the total weight of products that contain red meat (−321 g, 95% CI [−399 g, −242 g] and −421 g, 95% CI [−501 g, −342 g], respectively, from a baseline of 1,603 g in the control condition). There were no significant impacts on sodium purchased.

Of the 3,518 participants who completed the shopping task (i.e., checked out and did not time out of the shopping task), only 9 did not respond to the post-shopping task questionnaire. Average total spending was $31.68 for those who responded to the questionnaire and $32.09 for those who did not, suggesting similar shopping behaviors between the 2 groups.

**Table 2. Impact of warning label, tax, and combined warning label and tax on calories, saturated fat, and sodium purchased (n = 3,518).**

| | Control | Warning Label | Tax | Warning Label +Tax |
|---|---|---|---|---|
| | Mean (95% CI[a]) | Difference vs. control (95% CI) | Difference vs. control (95% CI) | Difference vs. control (95% CI) |
| **Total calories (kcal)** | 9,815.9 (9,613.0, 10,018.9) | 10,040.5[AB] (9,810.5, 10,270.4) | 9,632.5[A] (9,425.0, 9,840.1) | 9,504*[B] (9,315.1, 9,692.9) |
| **Total saturated fat (g)** | 155.5 (149.9, 161.0) | 155.8[AB] (150.3, 161.3) | 145.2*[A] (139.6, 150.7) | 142.7*[B] (137.1, 148.3) |
| **Total sodium (g)** | 18.2 (17.6, 18.8) | 18.0 (17.4, 18.6) | 17.6 (16.9, 18.2) | 17.9 (17.3, 18.5) |
| **Grams of products containing red meat (g)** | 1,603.3 (1,543.5, 1,663.2) | -78.5[AB] (-166.8, 9.8) | -320.5*[AC] (-399.0, -241.9) | -421.4*[BC] (-501.2, -341.6) |

[a] CI, confidence interval.

* Statistically significant differences with the control at the 5% level. P-values were calculated using linear regression models.

Shared superscripts A, B, and C indicate statistically significant differences between intervention arms at the 5% level.

**Table 3. Impact of warning label, tax, and combined warning label and tax on psychological outcomes [a].**

| | Control | Warning Label | Tax | Warning Label + Tax |
|---|---|---|---|---|
| | Mean (95% CI [b]) | Difference vs. control (95% CI) | Difference vs. control (95% CI) | Difference vs. control (95% CI) |
| **Healthfulness-related items** | | | | |
| Perceived healthfulness of eating red meat in general (n = 3,505) | 3.3 (3.2, 3.3) | -0.2*A (-0.3, -0.1) | 0.1AB (-0.0, 0.1) | -0.2*B (-0.3, -0.1) |
| Perceived healthfulness of specific red meat products (n = 3,498) | 2.9 (2.8, 2.9) | -0.2*A (-0.3, -0.1) | 0.0AB (-0.0, 0.1) | -0.2*B (-0.2, -0.1) |
| Perceived risk of cancer from eating red meat (n = 3,506) | 2.8 (2.8, 2.9) | 0.3*A (0.2, 0.3) | -0.1*AB (-0.2, -0.0) | 0.2*B (0.1, 0.3) |
| Thinking about the health harms of products while shopping (n = 3,505) | 2.7 (2.7, 2.8) | 0.1A (-0.0, 0.2) | -0.1AB (-0.2, 0.0) | 0.0B (-0.1, 0.1) |
| **Environment-related items** | | | | |
| Perceived environmental harm of eating red meat in general (n = 3,505) | 2.7 (2.6, 2.8) | 0.1*A (0.0, 0.3) | -0.1AB (-0.2, 0.0) | 0.1B (-0.0, 0.2) |
| Perceived sustainability of specific red meat products (n = 3,498) | 2.9 (2.9, 3.0) | -0.2*A (-0.3, -0.2) | 0.0AB (-0.1, 0.1) | -0.2*B (-0.3, -0.1) |
| Thinking about the environmental harm of products while shopping (n = 3,504) | 2.2 (2.2, 2.3) | 0.2*A (0.1, 0.3) | -0.1AB (-0.2, 0.0) | 0.1*B (0.0, 0.3) |
| **Cost-related items** | | | | |
| Perceived cost of burger product (n = 3,492) | 3.6 (3.6, 3.7) | 0.0AB (-0.1, 0.1) | 0.3*A (0.2, 0.4) | 0.2*B (0.1, 0.3) |
| Perceived cost of pizza product (n = 3,496) | 2.8 (2.7, 2.8) | 0.0AB (-0.1, 0.1) | 0.2*A (0.2, 0.3) | 0.3*B (0.2, 0.4) |
| Perceived cost of ham product (n = 3,494) | 3.2 (3.1, 3.3) | 0.0AB (-0.1, 0.1) | 0.3*A (0.2, 0.4) | 0.3*B (0.2, 0.4) |
| Thinking about the price of products while shopping (n = 3,507) | 4.1 (4.0, 4.2) | -0.0A (-0.1, 0.0) | 0.0A (-0.0, 0.1) | -0.0 (-0.1, 0.1) |
| **Red meat consumption intentions** | | | | |
| Intention to reduce red meat consumption in the next 30 days (n = 3,506) | 2.4 (2.3, 2.5) | 0.1A (-0.0, 0.2) | -0.1AB (-0.2, 0.0) | 0.1B (-0.1, 0.2) |

[a] All outcomes were measured on a scale of 1 to 5, with a higher value indicating a higher amount of the construct.

[b] CI, confidence interval.

* Statistically significant differences between the intervention and the control at the 5% level. P-values were calculated using linear regression models.

Shared superscripts A, B, and C indicate statistically significant differences between intervention arms at the 5% level.

Conditions that included a warning (i.e., the warning and combined conditions) decreased the perceived healthfulness and environmental sustainability of red meat. By contrast, policies that included a tax (i.e., the tax condition and combined condition) increased perceived cost (Table 3). No condition affected intentions to reduce red meat consumption in the next month.

In the control condition, policy support was low for taxes (21.4% of participants in favor and 55.7% against) and higher for health and environmental warning labels on red meat (42.1% and 40.2% of participants in favor and 28.5% and 32.2% against, respectively). Exposure to the tax led to lower support for taxes and higher opposition to taxes. Exposure to the taxes also led to lower support for environmental warning labels and increased opposition to health warning labels. In contrast, exposure to warning labels led to increased support for health and environmental warning labels; there was no effect on support for taxes or opposition for taxes or labels (S8 Table).

Process measures indicated the store was acceptable to participants. The majority reported that the online experimental store was easy to use, they could easily find all food and beverage items they were looking for, there were enough options, the store felt like a real online grocery store, and the items they selected in the experimental store were similar to their usual grocery purchases (**S1 Fig**).

## Sensitivity analyses

The main results for the primary outcomes were largely unchanged when we excluded participants in the bottom 2% of total expenditure, participants who completed the study in less than half the median time, or participants who purchased less than 5 of the product types on the shopping list (**S3 Appendix**).

Dropping out of the study was statistically significantly associated with several shopper characteristics. There were disproportionately more dropouts among older participants, participants doing less of their household's grocery shopping, and participants doing none of their grocery shopping online (**S9 Table**). However, Heckman selection models and inverse probability weighted regressions to account for differential dropout produced very similar results to the main model (**S10** and **S11 Tables**).

## Discussion

In a randomized trial, we found that both warning labels and a tax led to modest reductions in purchases of items containing red meat. The largest impact came from combining both warnings and the tax, resulting in an approximately 21% to 26% relative reduction of red meat purchases across outcomes. This reduction in purchases of items containing red meat translated into a reduction in calories and saturated fat of the shopping basket, but not sodium.

Our findings add to the growing body of evidence that taxes and warning labels applied to foods reduce purchases. Although limited research has explored the impact of these policies in the context of environmentally harmful products, our results are in line with nascent evidence that environment-focused fiscal and labelling policies may shift food purchasing behaviors. For example, a recent study by Wolfson and colleagues that found that, among US consumers, climate impact labels—and particularly negatively framed "high-impact" labels—led to an increase in sustainable selections from a fast food menu [30]. Our results showed that taxes also reduced red meat purchases and that these reductions were larger than for warnings, albeit by a small margin. These results are in line with a choice experiment conducted in Australia, which reported that price changes had a larger effect on selection of healthy and sustainable food options than did labels or logos [31]. These results are also in line with economic theory, which suggests that educational interventions alone will be less likely to lead to meaningful reductions in red meat consumption than a tax [32].

Warnings and taxes alone led to small reductions in red meat purchases, and the combined label and tax condition had the largest impact on red meat purchases. These results are consistent with a recent United Kingdom–based study in a simulated lunchtime canteen, which also found that labeling and taxes were most effective at shifting choice to meals with a lower carbon footprint when used in combination [33]. A Canadian study of sugar reduction policies also found additive effects on sugar purchases when taxes and front-of-package labels were implemented together [34]. One explanation is that these policies operate through distinct and complementary pathways: Warning labels can inform consumers and increase risk appraisals, whereas taxes incentivize consumers to select lower-cost (untaxed) items. Indeed, we also found that warning labels changed perceptions of the healthfulness and sustainability of red meat, whereas taxes influenced perceptions of cost. Together, these results suggest that the US

could achieve the greatest public health and environmental impact by implementing warnings and taxes together.

In addition to reducing red meat purchases, warning labels and taxes also reduced purchases of nutrients. Specifically, the combined tax and warning label condition led to reductions in both total energy and saturated fat, while taxes reduced saturated fat, and no condition impacted sodium. These results highlight the importance of understanding how these interventions affect substitutions across the food basket and ultimately influence the overall nutritional quality of diets. Moreover, a better understanding of substitutions is needed, as net health and environmental effects of policy actions to reduce red meat will vary depending on whether participants substituted to poultry, a plant-based meat alternative, beans, vegetables, or another food. Indeed, a recent simulation study in New Zealand found that while all meat replacement diets were associated with health gains and greenhouse gas reductions, replacement with minimally processed plant-based foods led to greatest benefits for all outcomes relative to ultraprocessed plant food alternatives, cellular meats, or diets based on other recommendations (e.g., EAT-*Lancet*) [35]. Future research will be critical for understanding how to design policies to not only discourage red meat intake but also promote healthy and sustainable substitutions.

We found that policies may have a differential impact for some subpopulations. For example, the combined tax and warning label condition led to a larger reduction in red meat purchases for younger versus older adults. One possibility is that younger people have greater knowledge of and concern about sustainability issues with regard to food [36,37] and thus are more responsive to policies. We also found some evidence of higher impact in the combined condition for lower-educated populations compared to those with a graduate degree. A recent study found that higher-educated US adults were more likely to report efforts to reduce red meat consumption [4], suggesting those with a graduate degree are already making efforts to reduce and may therefore be less likely to be further influenced by red meat policies. Other research has shown that education is associated with baseline knowledge about the health and environmental harms of eating red meat [17]; thus, policies such as warning labels, which seek to increase awareness, may have little impact among highly educated groups. The findings of a greater potential impact among participants with lower education suggests that such policies could promote health equity by leading to greater dietary change in lower educated groups, who tend to suffer a disproportionate burden of diet-related diseases, including cardiovascular disease [38–40] and cancer [41,42].

This study used both health and environmental warnings in the labeling policy condition, so we were unable to disentangle the effects of the 2 types of warnings. Our previous research has found that health warnings and "combined" warnings that include both health and environmental messages are perceived as more effective than warnings that include only environmental messages [24,25] suggesting that the health warning may have been responsible for most of the observed reductions in red meat purchases in this study. In contrast, a study in the UK found that nutrition labels (Nutriscore) did not impact the healthfulness of purchases in an online experimental store but that environmentally focused "ecolabels," either alone or in combination with nutrition labels, improved the sustainability of purchases [43]. Future research should examine whether health and environmental warnings have a differential impact on purchasing behaviors, and whether either type of warning is more likely to lead to reactance (message rejection) or label fatigue.

This study has strengths and limitations. One strength was the use of a naturalistic online grocery store designed to mimic a real-world US retailer. The use of these online stores provides a major advantage in that participants can choose from a wide assortment of real products in a setting that looks and feels like a real store. A recent validation study found that

expenditures in the naturalistic online grocery store used in this study were moderately-to-strongly correlated with expenditures made in a real online grocery store and that 95% of participants reported making similar purchases in the naturalistic online store as they did in a real online store [44]. However, in this study, it is unclear the extent to which aspects of this experiment (e.g., the use of a prespecified shopping list, lack of financial transactions) may have altered purchasing behaviors and attenuated or amplified the impact of the policies. For example, it is possible that in the real world, where habits, preferences, and other concomitant secular trends could influence consumer response, the effects of the policies could be attenuated. Or, real-world effects could be stronger because people are spending real money.

In addition, our study sample differed in sociodemographic characteristics from the US population (e.g., our sample included somewhat more women and respondents who identified as non-Hispanic white), though it is unclear whether this was due to the online panel, the fact that we included only meat consumers, or both. A related limitation is that, while online food retail is growing, only about 30% to 40% of Americans shop for groceries online [45,46]. It is not clear whether online food shopping behavior differs from in-person shopping behaviors, and, thus, it is unclear how generalizable our results are to consumers who do not shop online. Future research could address these limitations by testing these policies in a real-world brick-and-mortar retail setting where participants use their own shopping lists, spend their own money, and can take home products for consumption.

This study has several public health and policy implications. The results show that taxes and warning labels reduce red meat purchases among red meat consumers, suggesting that adoption and implementation of these policies could yield both health and environmental co-benefits. However, the likelihood of policy adoption is unclear. While several countries in Europe have recently considered adopting meat taxes [47–49], policy discussions on red meat in the US have been limited. A recent legal analysis found that while public policies to encourage meat reduction are legally feasible in the US, political feasibility is highly uncertain given the economic and political power of meat industry lobbies [50]. Policymakers may wish to consider the potential benefits to consumers, along with the legal and political feasibility, when weighing regulatory options for reducing intake of red and processed meat. Another barrier to policy adoption is public support: In this study, overall support for both policies was low (<50% of participants), and similar to a previous study [51], support was lower for taxation than labeling. However, some evidence suggests that this barrier could be overcome through different types of messaging about the policy. For example, one study in Germany found that consumers were more likely to support a meat tax if it was justified by animal welfare than by climate change mitigation [52]. Research will be necessary to understand what type of framing or justification is likely to elicit higher support for meat reduction policies in the US.

## Conclusions

In a randomized controlled trial using a naturalistic online grocery store, we found that taxes and warning labels reduced purchases of red meat purchases as well as calories and saturated fat. The condition that included both taxes and warning labels had the biggest impact, suggesting this combined approach would achieve the greatest co-benefits for health and environment.

## Supporting information

**S1 CONSORT Checklist. Reporting checklist for randomised trial.**
(DOCX)

**S1 Appendix. Experimental setting.**
(DOCX)

**S2 Appendix. Moderator operationalization.**
(DOCX)

**S3 Appendix. Sensitivity analyses results.**
(DOCX)

**S1 Table. Survey measures used in the randomized trial.**
(DOCX)

**S2 Table. Comparison of sample statistics with population statistics.** [a]Sources: Decennial Census 2020 for age, sex, race, and ethnicity and American Community Survey 2021 for household income in the past 12 months. [b]Participants who completed the shopping task ($n$ = 3,518). [c]Sample: Sex refers to gender (sex information not collected and population statistics on gender not found). [d]Alone or in combination with other races. [e]Our questionnaire included being of Spanish origin together with Hispanic or Latino ethnicity.
(DOCX)

**S3 Table. Primary outcome descriptive statistics by trial condition ($n$ = 3,518).** [a]SD, standard deviation.
(DOCX)

**S4 Table. Secondary outcome descriptive statistics by trial condition.** [a]All outcomes except total calories, saturated fat, and sodium were measured on a scale of 1 to 5, with a higher value indicating a higher amount of the construct. [b]SD, standard deviation.
(DOCX)

**S5 Table. Difference in primary outcomes by trial condition ($n$ = 3,518).** [*]Bold denotes statistically significant differences between each intervention and the control at the 5% level. [a]Difference is control compared to intervention. [b]Shared superscript indicates significant difference between interventions at the 5% level. [c]CI, confidence interval. [d]$P$ values calculated using fractional probit and Poisson regression models for percent and count of red meat products, respectively.
(DOCX)

**S6 Table. Moderation results for impact of warning labels, tax, and combined warning label + tax interventions on count of red meat purchased by demographic characteristics.** [a]Difference is control compared to intervention. [b]Wald test of equal differences with the control across the moderator's levels. [c]CI, confidence interval. [d]Based on a scale of self-perceived dietary behavior [28]. [e]Based on the GREEN Scale [29].
(DOCX)

**S7 Table. Moderation results for impact of warning labels, tax, and combined warning label + tax interventions on percentage of red meat purchased by demographic characteristics.** [a]Difference is control compared to intervention. [b]Wald test of equal differences with the control across the moderator's levels. [c]CI, confidence interval. [d]Based on a scale of self-perceived dietary behavior [28]. [e]Based on the GREEN Scale [29].
(DOCX)

**S8 Table. Differences in policy support and opposition by trial arm ($n$ = 3,502).** [*]Asterisk indicates statistically significant differences between each intervention and the control at the 5% level. [a]Shared superscript indicates a statistically significant difference between

interventions at the 5% level. [b]CI, confidence interval.
(DOCX)

**S9 Table. Screener characteristics by shopping task completion status among participants who started the shopping task ($n$ = 4,106).** [a]Pearson's $\chi^2$ (chi-squared) test. [b]Expected counts (rounded to the nearest integer) in parentheses (only shown if $p < 0.05$).
(DOCX)

**S10 Table. Summary statistics for the inverse probability weights in the final sample ($n$ = 3,518).**
(DOCX)

**S11 Table. Sensitivity analyses results.** [*]Asterisk denotes statistically significant differences between each intervention and the control at the 5% level. [a]Superscripts indicate pairwise differences between intervention conditions at the 5% level. [b]CI, confidence interval. [c]P values calculated using Poisson and fractional probit regression models for the count and percent of red meat items. [d]The median completion time (shopping task and questionnaire) was 18 minutes and 46 seconds. [e]All other participants (i.e., participants who purchased at least 5 of the 9 shopping list items and potentially other items not on the shopping list) were included.
(DOCX)

**S1 Fig. Process measures.** [*]$n$ = 3,508; [**]$n$ = 3,507; [***]$n$ = 3,509.
(TIF)

**S1 Statistical Analysis Plan. Developing and Evaluating Product Messaging Hypotheses and Analytic Plan.**
(PDF)

**S1 Study Protocol. Developing and Evaluating Product Messaging.**
(PDF)

## Acknowledgments

The authors thank Pasquale Rummo for assisting with the development of the online grocery store. The authors thank David Gonzalez, Jamie Halula, Sophia Hurr, Keilei Latragna, Rhea Naik, and Carla Seet for their help with cleaning the dataset and reviewing the online grocery store. They would also like to thank Dhara Buebel for assisting with the manuscript preparation, Emily Busey for designing the online grocery store logo, Christina Chauvenet for assistance with project management, Bridget Hollingsworth for providing guidance on tagging red meat products, Yiqing Zhang for providing expertise on the data scraping process, and Wilma Waterlander for providing general feedback for the online store development process. Finally, the authors thank Susan Jebb and Brian Cook for contributing their expertise from the development of the Woods supermarket platform.

## Author Contributions

**Conceptualization:** Lindsey Smith Taillie, Maxime Bercholz, Carmen E. Prestemon, Isabella C. A. Higgins, Anna H. Grummon, Marissa G. Hall, Lindsay M. Jaacks.

**Data curation:** Maxime Bercholz, Carmen E. Prestemon.

**Formal analysis:** Maxime Bercholz.

**Funding acquisition:** Lindsey Smith Taillie, Lindsay M. Jaacks.

**Project administration:** Carmen E. Prestemon, Isabella C. A. Higgins.

**Writing – original draft:** Lindsey Smith Taillie, Maxime Bercholz, Carmen E. Prestemon, Lindsay M. Jaacks.

**Writing – review & editing:** Isabella C. A. Higgins, Anna H. Grummon, Marissa G. Hall.

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
