## [Editor Report · Decision Letter 0]

16 Apr 2023

Dear Dr Taillie, 

Thank you for submitting your manuscript entitled "Impact of taxes and warning labels on red meat purchases in a naturalistic online grocery store: A randomized controlled trial" for consideration by PLOS Medicine.

Your manuscript has now been evaluated by the PLOS Medicine editorial staff and I am writing to let you know that we would like to send your submission out for external assessment.

However, before we can send your manuscript to reviewers, we need you to complete your submission by providing the metadata that are required for full assessment. To this end, please login to Editorial Manager where you will find the paper in the 'Submissions Needing Revisions' folder on your homepage. Please click 'Revise Submission' from the Action Links and complete all additional questions in the submission questionnaire.

Please re-submit your manuscript within two working days, i.e. by Apr 18 2023 11:59PM.

Once your full submission is complete, your paper will undergo a series of checks in preparation for external assessment. 

Kind regards,

Richard Turner PhD

Consulting Editor, PLOS Medicine

plosmedicine@plos.org

---

## [Decision Letter · Decision Letter 1]

16 May 2023

Dear Dr. Taillie,

Thank you very much for submitting your manuscript "Impact of taxes and warning labels on red meat purchases in a naturalistic online grocery store: A randomized controlled trial" (PMEDICINE-D-23-00922R1) for consideration at PLOS Medicine. 

[LINK]

In light of these reviews, we will not be able to accept the manuscript for publication in the journal in its current form, but we would like to consider a revised version that addresses the reviewers' and editors' comments. We cannot make any decision about publication until we have seen the revised manuscript and your response, and we plan to seek re-review by one or more of the reviewers. 

We expect to receive your revised manuscript by Jun 06 2023 11:59PM. Please email us (plosmedicine@plos.org) if you have any questions or concerns.

We look forward to receiving your revised manuscript. 

Sincerely,

Katrien Janin, 

PLOS Medicine

plosmedicine.org

Notes from the Academic Editor:

It is an interesting paper on a really important topic. A few comments:

1. The intro says few studies have been conducted on warning labels and taxes for meat. But there are some and I think it would be appropriate to review this literature briefly in the intro to indicate what's already known on the topic. eg. https://bmcpublichealth.biomedcentral.com/articles/10.1186/s12889-020-08590-z and https://journals.plos.org/plosone/article?id=10.1371/journal.pone.0204139 Similarly, I'd like to see some of the theory around why these particular interventions - beyond simply that they've been effective in other settings - described here. Some of this is information is already in the discussion, but to my mind it is useful for the reader to have that background when considering this place of this new work as they read through it.

2. I think the sample is convenience, not purposive. If the latter, they need to be clear what purpose trying to achieve.

3. Whilst I appreciate the substantial work put into this experimental naturalistic online grocery store, it remains an experimental setting. There is some mild indication in the discussion that behaviour in this may not be generalisable to real-life, but hardly any consideration of this point. It would be great to see some data on how realistic behaviour in this environment is. If that's not available, then that needs to be said. Key issues include: a minority of the population use online grocery shopping regularly so however naturalistic it is, this is not an everyday experience for them; they aren't spending their own money; they aren't responding to their 'normal' grocery shopping needs. For this reason, I find it quite hard to judge how important the findings might be.

4. In terms of novelty, I'm not sure it's a major advance to show that the effect of taxes and warning labels generalise to foods they haven't explicitly commonly used previously - the paper could justify why they may or may not to help us understand this. But it is an important thing to confirm.

Your manuscript has been assessed by three reviewers whose reports can be found below. As you will see from the comments, the reviewers have raised a number of concerns that need addressing. Additionally, the academic editor has also provided comments. Please carefully revise the manuscript to address all comments raised.

CLINICAL TRIALS STUDY

In accordance with ICMJE requirements, PLOS Medicine requires prospective, public registration of a data sharing plan (as part of mandatory clinical trials registration) for all clinical trials that began enrollment on or after January 1, 2019.

Please include a CONSORT flow chart. In the flow diagram, please indicate the number of individuals in each group analyzed in the ITT analysis. We note you currently have the flow chart as Supplementary Figure 1.; Please consider it moving to the main manuscript.

Please include the study protocol document and analysis plan, with any amendments, as Supporting Information to be published with the manuscript if accepted.

DATA AVAILABILITY STATEMENT

Thank you for including a statement that the data is available without restriction on the Harvard Dataverse database. Please include the specific link to where the data can be found.

ABSTRACT:

Please structure your abstract using the PLOS Medicine headings: Background, Methods and Findings, Conclusions. 

Please remove all other subheaders.

Abstract Background: 

The final sentence should clearly state the study question.

ABSTRACT - Methods and Findings:

Please provide the number in each group

Please indicate the participants enrollment date(s)

Please quantify the main results (with 95% CIs and p values).

In the last sentence of the Abstract Methods and Findings section, please describe the main limitation(s) of the study's methodology.

AUTHORS SUMMARY

Ideally each sub-heading should contain 2-3 single sentence, concise bullet points containing the most salient points from your study.

In the final bullet point of ‘What Do These Findings Mean?’, please include the main limitations of the study in non-technical language.

INTRODUCTION

Please conclude the Introduction with a clear description of the study question (which you have done) and the approach taken to address it.

METHODS and RESULTS

Please specify whether informed consent was written or oral (line 94)

Please include the number of participants in each group (see design line 94-95).

Line 191: “… ANOVA to detect… “ Please define all abbreviations at first use, check and amend throughout. This includes reporting statistical information abbreviations.

 please define all abbreviations at first use. 

Please report p values as p<0.001 and where higher as p=0.002 (e.g. see line 254: the warning label condition (p<0.01)).

Please present the primary and secondary outcomes of the study in the main paper (not in the Supporting Information files).

DISCUSSION

Please present and organize the Discussion as follows: a short, clear summary of the article's findings; what the study adds to existing research and where and why the results may differ from previous research; strengths and limitations of the study; implications and next steps for research, clinical practice, and/or public policy; one-paragraph conclusion.

Line 300-301: “To our knowledge, this was the first study to test taxes and warning labels on red meat purchases in a naturalistic online grocery store setting among US consumers”. 

Please avoid assertions of primacy. 

ACKNOWLEDGMENTS/ DECLARATIONS

Please remove all statements apart from acknowledgements, author contributions and abbreviations from the end of the main manuscript and include these only in the relevant parts of the manuscript submission form. Funding, competing interest, and data availability will be compiled as metadata.

REFERENCES

Please use the "Vancouver" style for reference formatting, and see our website for other reference guidelines https://journals.plos.org/plosmedicine/s/submission-guidelines#loc-references

Please ensure that in the bibliography up to but no more than 6 author names are listed, followed by et al., in the event that more than 6 authors contribute to an individual study. Journal name abbreviations should be those found in the National Center for Biotechnology Information (NCBI) databases.

FIGURES

Figure 1: We note you have included images of branded (copyrighted) products. Please confirm you have appropriate permission or you may wish to remove these images. For more information, please see our guidelines: https://journals.plos.org/plosmedicine/s/figures#loc-licenses-and-copyright and https://journals.plos.org/plosmedicine/s/licenses-and-copyright

TABLES

Please ensure that any and all abbreviations detailed in the tables are clearly defined in each caption/legend for the reader (e.g. see CI, for table 2,3)

SUPPORTING INFORMATION

Figures: we note you have included images of branded products. Please remove these unless you can provide permission. For more information see https://journals.plos.org/plosmedicine/s/figures#loc-licenses-and-copyright and https://journals.plos.org/plosmedicine/s/licenses-and-copyright

Please ensure that any and all abbreviations detailed in the tables and figures are clearly defined in each caption/legend for the reader (e.g Supplemental Table 4 and 7: CI, Supplemental Table 5: SD, etc. )

Please consider moving supplementary figure 1: the participant flow chart to the main manuscript.

Comments from the reviewers:

Reviewer #1: This study aims to test the impact of taxes and warning labels on red meat purchases in the US.

Comments:

"A purposive sample of 5,533 participants was recruited in October 2021 by CloudResearch (Prime Research Solutions LLC, New York) to complete an online survey and experiment. CloudResearch targeted recruitment to create a sample that approximately represented the US population in terms of gender, race/ethnicity, income, and age."

Can the authors please assess whether the final analysed cohort can be considered to be representative of the wider population, for generalisability of the study findings?

"After providing informed consent, participants were randomly assigned to one of four conditions in a 1:1:1:1 ratio: control, warning labels, tax, or combined (warning labels + taxes)."

The authors have followed a rigorous study design, helping to reduce possible sources of bias and confounding in the data and analysis.

"Participants completed a shopping task in the naturalistic online grocery store. Participants selected items using a pre-determined nine-item shopping list: 1 pizza, 1 burrito, burger patties (meat or vegetarian), breakfast sausages (meat or vegetarian), 1 frozen individual meal, 1 loaf of bread, 1 sandwich filling (for example, ham, turkey, or peanut butter), 1 pack of tortillas, and 1 taco filling (for example, steak, chicken, or beans). The shopping list was informed by data on the main sources of red meat consumption in the United States."

and

"Participants were informed that their budget was $40"

The participants completed the task under the same conditions as each other, which helps to reduce potential confounding. However, can the authors please comment on whether this may also restrict the generalisability of the study findings (i.e. the structure of the study means the research question is conditional on the shopping list and budget)?

"To encourage participants to select products that they would purchase in real life, participants were informed that 10% of participants would be randomly selected to receive the groceries they selected during the shopping task and the remainder of their budget in cash. In reality, 10% of participants were randomly selected to receive a $40 gift card."

The approach taken by the authors seems sensible, and helps to promote more realistic shopping tendencies of the participants.

"After completing the shopping task, participants were directed to a survey to complete a post-shopping questionnaire. All participants were given an incentive paid in points redeemable by Cloud Research for completion of the survey and experiment. "

Can the authors please clarify how many of the shopping participants went on to complete this survey?

Can the authors please explore whether the particpants of the survey were more likely to be the participants who spent less money on the shop (e.g. were those with a tendency to save money in the shop, more prone to respond to an incentive for participation in the survey)?

"For policy support outcomes, for simplicity of interpretation, we created a 160 binary variable, 'agree' or 'strongly agree' vs. the remaining three response options ('neither agree nor disagree', 'disagree', and 'strongly disagree')."

Did the authors consider conducting a sensitivity analysis on this dichotomisation, including 'neither agree nor disagree' in the alternative categorisation for example?

"Analyses and hypotheses were pre-registered on http://www.clinicaltrials.gov (NCT04716010). "

Can the authors please supply a copy of the study protocol and statistical analysis plan (SAP) in the next submission?

"Per CONSORT guidelines, we did not test for balance in covariates across conditions"

The authors have suitably provided the completed CONSORT checklist in the supplementary material.

"The study was originally powered for an ANOVA to detect a Cohen's f=0.05 main effect of the warning labels with 80% power at the 5% level (n=786 per condition). However, we chose to compare individual conditions instead, and post-hoc power calculations showed that we had a large enough sample (n=866 to 891 per condition) to detect a Cohen's d=0.13 difference between the warnings and control conditions at the 5% level with 80% power."

The authors transparently communicate the basis and amendment to the sample size calculations.

"We used Poisson and fractional probit regression models for the count and fraction of items in the basket that contained red meat, respectively, and linear regression models for the remaining outcomes, with condition as the independent variable. We used robust standard errors in all models except linear regression models where we did not find evidence of heteroskedastic errors based on Breusch-Pagan test assuming IID errors. Differences between conditions were assessed by comparing the predicted means by condition."

and

"For each of these potential moderators, we fit a Poisson regression model for the count outcome and a fractional probit model for the fractional outcome with condition, the moderator, and their interaction as the independent variables. "

The authors have applied technically appropriate statistical methods.

"Using the model estimates, we then estimated the difference between each intervention condition and the control condition at each level of the moderator and tested the hypothesis of equal differences with the control condition across those levels for each intervention condition, examining all pairwise comparisons (e.g., for age, 18- 39y vs. 40-59y, 18-39y vs. 60y or over, and 40-59y vs. 60y or over)."

Did the authors consider correcting for multiple comparions here?

"As pre-registered, we carried out sensitivity analyses for the main comparisons on the primary outcomes. "

The authors have conducted some insightful additional analyses which help to demonstrate the robustness of the study findings.

Overall, the Results are presented accurately and the Discussion is well balanced. 

"One limitation of this study is that the intervention focused only on red meat, rather than aiming to reduce both red and processed meat. Because Americans have high consumption of processed meat and consumption of processed meat is linked to an array of adverse health effects, future studies should examine how policies impact both red and processed meat. ... At the same time, it is possible that effects of these policies could be attenuated in the real world, where habits, preferences, and other concomitant secular trends (e.g., price inflation) could influence consumer response." 

Can the authors please expand on the discussion of the study limitations?

Reviewer #2: Thank you for the opportunity to review this paper which makes a strong and original contribution to the field. How best to proceed with public health interventions on food and drink consumption is a live issue and this paper provides additional robust evidence on the relative benefits of warning labelling and taxation. This is of relevance to policymakers in particular, but also of interest to clinicians experiencing the downstream effects of (lack of) policy.

A few comments for consideration by the author:

Data availability - Please ensure that a DOI or relevant identifier is included for the data files prior to publication.

Charts - It may be worth using a different signifier for significant differences as bold text is quite difficult to see, at least at the resolution used in the paper.

Line 57 - Please check citation 3. The percentage cited appears to be in the main report but not the appendix.

Lines 66-68 - Please consider adding here or in the discussion section a few further references to studies on public acceptability on labelling and taxation (https://doi.org/10.1016/j.socscimed.2019.112395;
https://doi.org/10.1038/s43016-023-00696-y); cost-effectiveness of such policies (https://doi.org/10.1016/j.amepre.2019.02.023); simulations of health impact (https://doi.org/10.3390/nu15041020); development of economic theory on the subject (https://doi.org/10.1002/ajae.12016;
https://doi.org/10.1086/721078); and relevant policy developments (https://www.weforum.org/agenda/2019/08/meat-tax-denmark-sweden-and-germany/;
https://www.fas.usda.gov/data/netherlands-concept-meat-tax-under-discussion-netherlands).

Discussion - An additional point that is worth including is the combination of environmental and health warning labels in the study. Future research should separate these out and examine whether either is more effective and whether including more than one has diminishing returns or unintended consequences of creating awareness 'fatigue' (leading to a reduced response).

Reviewer #3: This manuscript describes an experiment to test the impact of taxes and warning labels on red meat purchases in a virtual (naturalistic) online grocery store. The warning label, 30% tax, and combined warning + tax conditions led to modest reductions in red meat-containing items purchased. The randomized controlled trial design is appropriate and reporting of the study methods and findings was clear and complete. Below are suggestions to increase clarity in places:

Lines 71-72: Secondary objectives included testing the impact of warnings and taxes on psychological measures and support for policies but these were not reported in the abstract. For completeness, I suggest these results are also reported briefly in the abstract.

Line 79: Can the authors comment on how representative the 13,000 products in their naturalistic online grocery store were of the full range of products stocked by the grocery retailer they set out to mirror? It would be useful to know if they included products from all departments and aisles or just a selection and, if the latter, how selections were made.

Line 85: A purposive sample of 5533 participants was recruited and intended to represent the US population in terms of gender, race/ethnicity, income, and age. However, only 3518 of these (63.6%) completed the study (line 228). The authors should describe how the final sample differed from the intended sample in terms of key demographics and if the final sample was representative of the US population in terms of gender, race/ethnicity, income, and age. Sample demographics are presented in a table but there is no comment on their representativeness. I note that the overall sample was 60% women, 43% college-educated and 73% white so it seems unlikely that it represents the diversity of the US population. this has implications for the generalisability of the study findings to other population groups.

Line 157: The authors should explain the significance of Cronbach's alpha being greater or less than 0.7 (and hence why outcomes were combined or analyzed separately).

Lines 188-194: What is the clinical significance of a Cohen's f = 0.05 or Cohen's d = 0.13? It would be good to relate these statistics to tangible differences in red meat purchases if possible.

Lines 205-215: Are moderation analyses appropriate when reporting RCT results? The randomized nature of the experiment means personal characteristics should be balanced across groups so I am unsure of the benefit (or indeed appropriateness) of moderation analyses with this study design. They are not usually reported for large public health interventions as far as I am aware.

[LINK]

---

## [Decision Letter · Decision Letter 2]

26 Jul 2023

Dear Dr. Taillie,

Thank you very much for re-submitting your manuscript "Impact of taxes and warning labels on red meat purchases in a naturalistic online grocery store: A randomized controlled trial" (PMEDICINE-D-23-00922R2) for review by PLOS Medicine.

I have discussed the paper with my colleagues and the academic editor and it was also seen again by two reviewers. I am pleased to say that provided the remaining editorial and production issues are dealt with we are planning to accept the paper for publication in the journal.

[LINK]

We look forward to receiving the revised manuscript by Aug 02 2023 11:59PM.   

Sincerely,

Katrien Janin, PhD

Senior Editor 

PLOS Medicine

plosmedicine.org

Requests from Editors:

As you can see below, the reviewers and the AE are satisfied with the responses and revisions you made to the manuscript.

We do have some minor editorial comments:

CT Studies

Thank you for providing your CONSORT checklist. Please replace the page numbers with paragraph numbers per section (e.g. "Methods, paragraph 1"), since the page numbers of the final published paper may be different from the page numbers in the current manuscript. With my apologies for not having mentioned this in the previous revision round.

In accordance with ICMJE requirements, PLOS Medicine requires prospective, public

registration of a data sharing plan (as part of mandatory clinical trials registration) for all

clinical trials that began enrollment on or after January 1, 2019. Please confirm the registration of a data sharing plan and add link or add as SI. (Thank you for already adding the Statistical Analysis Plan and Study Protocol).

TITLE

Please add US or similar to your title to provide the locality context of your study.

ABSTRACT

For the abstract, please structure the abstracts along the following 3 headings: Background, Methods and Findings, Conclusions. (Currently there are separate headings for Methods and Findings sections). Please remove all other subheaders.

Please quote aggregate participant demographic characteristics in the abstract; please quote effect size + 95% CI for the primary outcome(s) - not just a p value. For p values, please report p values as p<0.001 and where higher please provide the exact number e.g. as p=0.002 (in line with the format used throughout the manuscript). 

Overall, please check and ensure you adhere to the CONSORT abstract checklist (for more details see here: https://www.equator-network.org/reporting-guidelines/consort-abstracts/). 

Last but not least, please add the trial registration number after the abstract. 

ETHICS STATEMENT

Please add consent information to the Ethics Statement

DATA AVAILABILITY

Please check if the provided URL is correct: https://doi.org/10.7910/DVN/HQMFUS

Currently, I receive an error message when trying to access it (DOI cannot be found) although I appreciate you have indicated that this URL will be active upon accept.

AUTHOR’S SUMMARY

In the final bullet point of ‘What Do These Findings Mean?’, please include the main limitations of the study in non-technical language.

RESULTS

E.g see lines 281 - 282: Suggest reporting statistical information as follows for clarity for the reader “…36%; (95% CI [35%,37%] p=0.001), instead of 36% [(35, 37), p=0.001].

Please provide 95% CIs and p values for all results were appropriate, check and amend throughout.

Please specify the statistical test used to determine p value.

SUPPORTING INFORMATION

Can you please re-check figure 3, error bars look identical (which can be correct but maybe have a look if indeed correct).

I note that S1 Table.doxc still contains branded images - see Survey measures appendix – Sample images. 

As before, please confirm you have appropriate permission or you may wish to remove these images. For more information, please see our guidelines: https://journals.plos.org/plosmedicine/s/figures#loc-licenses-and-copyright and https://journals.plos.org/plosmedicine/s/licenses-and-copyright

Please check and amend throughout, or provide permission.

SOCIAL MEDIA

To help us extend the reach of your research, if you wish please provide any Twitter handle(s) that would be appropriate to tag, including your own, your coauthors’, your institution, funder, or lab. Please respond to this email with any handles you wish to be included if and when we tweet about this paper.

Comments from Reviewers:

Reviewer #1: Many thanks to the authors for satisfactorily considering and responding to each comment in turn, amending the article as required.

Reviewer #2: Many thanks for responding to reviewer comments clearly and comprehensively. My concerns have been addressed in full and I am very happy to recommend publication. Thank you for the opportunity to review this important work.

[LINK]

---

## [Editor Report · Decision Letter 3]

22 Aug 2023

Dear Dr Taillie, 

On behalf of my colleagues and the Academic Editor, [AE Name], I am pleased to inform you that we have agreed to publish your manuscript "Impact of taxes and warning labels on red meat purchases among US consumers: A randomized controlled trial" (PMEDICINE-D-23-00922R3) in PLOS Medicine.

We have also added a final acceptance request: to remove the brande images from the SI. I checked with our copyright team and while we appreciate you blurred the brand names but the entire packaging is part of the brand image and thus may fall under a license.

PRESS

Sincerely, 

Katrien Janin, PhD

Senior Editor 

PLOS Medicine